# From Extraction to Advanced Analytical Methods: The Challenges of Melanin Analysis

**DOI:** 10.3390/ijms20163943

**Published:** 2019-08-13

**Authors:** Ioana-Ecaterina Pralea, Radu-Cristian Moldovan, Alina-Maria Petrache, Maria Ilieș, Simona-Codruța Hegheș, Irina Ielciu, Raul Nicoară, Mirela Moldovan, Mihaela Ene, Mihai Radu, Alina Uifălean, Cristina-Adela Iuga

**Affiliations:** 1MedFuture—Research Center for Advanced Medicine, “Iuliu Hațieganu” University of Medicine and Pharmacy, Louis Pasteur Street 4-6, Gh. Marinescu Street 23, 400349 Cluj-Napoca, Romania; 2Department of Pharmaceutical Analysis, Faculty of Pharmacy, “Iuliu Hațieganu” University of Medicine and Pharmacy, Louis Pasteur Street 6, 400349 Cluj-Napoca, Romania; 3Department of Pharmaceutical Botany, Faculty of Pharmacy, “Iuliu Hațieganu” University of Medicine and Pharmacy, Louis Pasteur Street 6, 400337 Cluj-Napoca, Romania; 4Department of Dermopharmacy and Cosmetics, Faculty of Pharmacy, ”Iuliu Hațieganu” University of Medicine and Pharmacy, Ion Creangă Street 12, 400010 Cluj-Napoca, Romania; 5Horia Hulubei National Institute for Physics and Nuclear Engineering, Reactorului Street 30, 077125 Măgurele, Romania

**Keywords:** melanin, eumelanin, pheomelanin, allomelanin, biopolymers, analytical methods, MALDI, pyrolysis gas chromatography

## Abstract

The generic term “melanin“ describes a black pigment of biological origin, although some melanins can be brown or even yellow. The pigment is characterized as a heterogenic polymer of phenolic or indolic nature, and the classification of eu-, pheo- and allo- melanin is broadly accepted. This classification is based on the chemical composition of the monomer subunit structure of the pigment. Due to the high heterogeneity of melanins, their analytical characterization can be a challenging task. In the present work, we synthesized the current information about the analytical methods which can be applied in melanin analysis workflow, from extraction and purification to high-throughput methods, such as matrix-assisted laser desorption/ionization mass-spectrometry or pyrolysis gas chromatography. Our thorough comparative evaluation of analytical data published so far on melanin analysis has proven to be a difficult task in terms of finding equivalent results, even when the same matrix was used. Moreover, we emphasize the importance of prior knowledge of melanin types and properties in order to select a valid experimental design using analytical methods that are able to deliver reliable results and draw consistent conclusions.

## 1. Introduction

Melanins, one of the most widespread pigments in the animal kingdom, represent a group of biopolymers with distinctive physicochemical and biological properties. However, their production is not only confined to animals, as melanins can be widely found in fungi, plants, or bacteria. 

The significant research interest in studying melanins is justified by their multiple functions. Most melanins harbor a unique capacity to interact with a wide range of electromagnetic radiation frequencies, thereby functioning as a protecting and energy harvesting pigment. The protective mechanisms of melanin imply changes in melanin’s chemical composition and structure, inelastic scattering of photons, non-radiative dissipation of absorbed photons and free-radical scavenging [1]. Moreover, melanin is able to effectively chelate metal ions, protecting cells from potentially toxic metal ions. As most of the radiation energy from sunlight is converted into heat, melanin also plays an important role in the thermoregulation of melanized organisms. Also, in pathogenic bacteria and fungi, melanin is a determinant virulence factor, acting as a shield against immunological host response [2].

As a consequence, melanin gained a commercial importance, manufacturers designing melanin based thin films for organic electronics and bioelectronics, functional nanoparticles and biointerfaces, sunscreens or environmental remediation devices [3].

The ubiquitous sources of melanin could be responsible for the its heterogenous structural features, as it is synthesized by the oxidative polymerization of different types of phenolic or indolic monomer units. Broadly, melanins can be categorized in eumelanin (dark brown/black), pheomelanin (yellow/red) and allomelanin, specific to the plant kingdom. Both eumelanin and pheomelanin derive from the common precursor dopaquinone, which is obtained through tyrosine or L-Dopa oxidation (Figure 1). 

Cyclization of dopaquinone results in cyclodopa which is rapidly oxidized to dopachrome. Dopachrome is converted to 5,6-dihydroxyindole (DHI) and 5,6-dihydroxyindole-2-carboxylic acid (DHICA) units which undergo oxidative polymerization to form eumelanin (Figure 1 and Figure 2a). Pheomelanin is obtained by oxidation of 5-S- and 2-S-cysteinyldopa units via benzothiazine and benzothiazole intermediates (Figure 1 and Figure 2b) [4,5]. y.

As a mixture of eumelanin and pheomelanin, neuromelanin is the only melanin pigment which is not formed in melanocytes, but rather in catecholaminergic neurons of the *substantia nigra* and in lower proportion in the *locus coeruleus* [2]. Neuromelanin has been proved to contain both benzothiazine and indole units, with cysteine in its pheomelanin core and eumelanin at the surface [6]. Allomelanins, a very heterogenous group, are nitrogen-free polymers which include pyomelanin, which is obtained through autooxidation and polymerization of homogentisic acid [2]. 

Beside these melanins, there are also synthetic melanins which are frequently used as model melanin for biophysical studies. Synthetic melanins are formed by chemical oxidation of dopa and dopamine precursors and they are not attached to any protein [2].

Fungal melanin, which is of great biotechnological and biomedical interest, bears specific molecular features. Most ascomycetes synthetize melanin through polyketide pathway, using 1,8-dihydroxynaphthalene (1,8-DHN) as precursor. Conversely, basidiomycetes and other imperfect fungi use alternative pathways, generally using L-3,4-dihyroxyphenylalanine (L-DOPA) as a precursor in a pathway which resembles mammalian melanin biosynthesis [2,7].

Despite melanin ubiquity and multifunctionality, there are many queries pending for clarification, especially regarding its molecular structure. Chemical characterization of melanin can be a challenging task as the pigment is highly heterogeneous, insoluble in organic solvents, hydrophobic, and resistant to chemical degradation. These physicochemical properties restrict the number of analytical approaches capable of identifying and characterizing the pigment. Moreover, data published so far is relatively inconsistent, since discrepancies in extraction and purification, sample preparation or data processing make results difficult to compare. On the other hand, applying the same procedures on different melanin sources, disregarding its cellular localization, its biosynthetic pathways or the possible interferences cannot provide equivalent results. 

As an example, clear differences in synthetic DHI and DHICA melanins exists, the former being a weak H-atom donor and a poor free radical scavenger, while the latter is a strong H-atom donor and a strong free radical scavenger. These are consequences of the different electron delocalization, absorption and paramagnetic properties, as well as the aggregation mode at the supramolecular level [3]. Based on these intrinsic differences, it is essential for analysts to take informed decisions when designing the workflow for melanin analysis. 

The present paper aims to provide relevant information regarding the most commonly applied methods in melanin analysis, from extraction and purification to high-throughput methods such as matrix-assisted laser desorption/ionization mass-spectrometry or pyrolysis gas chromatography. The overall purpose is to guide specialists in selecting the proper analytical methods for melanin analysis in order to correlate the results with existing literature and draw reliable conclusions. 

## 2. Extraction and Purification

The selection of melanin extraction and purification procedures is highly dependent on the melanin source (fungal, bacterial, human hair), localization (intracellular/extracellular), and the final purpose of the study.

As melanins are often tightly bound to other cellular components, isolation of pure melanin can be a challenging task. Most melanin extractions involve alkaline treatment of matrix with 1 M NaOH [8,9,10], 1 M KOH or 0.5 M NH_4_OH [11] by reflux in water bath or autoclavation. For human and mouse hair melanin, alkaline extraction is not suitable due to the insolubility of eumelanin. 

The destruction of the close association of proteins and other biological components with melanin can be achieved using boiling acids or bases for several hours, followed by successive washing steps of the precipitate. Melanin purification is usually accomplished by acid hydrolysis or successive washing steps with organic solvents such as chloroform, petroleum ether, ethyl acetate, acetone, or absolute ethanol [8,9,12,13,14,15]. Although numerous studies have used harsh hydrolytic extractions, studies have shown that melanins are decomposed to a significant extent on acid treatment, especially through extensive decarboxylation [16].

Therefore, milder methods have been developed, such as enzymatic procedures which use specific cell wall lysing enzymes, guanidine thiocyanate for protein denaturation and a serine proteinase for cleavage [17,18]. As an improvement, tissue grinding in liquid nitrogen prior to enzymatic digestion has been proposed in order to reduce the heat generated during the mechanical disruption [19]. 

Liu et al. made a detailed comparison of the morphology and spectroscopic properties of human hair melanin after acid/base and enzymatic extraction procedures [20]. Their study concluded that acid/base extraction alters the melanin molecular structure and affects the concentration and distribution of coordinated metals. Therefore, acid/base extracted melanin can be considered a poor model for the human pigment. In contrast, the enzymatically extracted melanin retains the morphology of intact melanosomes and it should be preferred when studying human melanin. 

Table 1 illustrates several extraction and purification steps proposed for various melanin types. Depending on melanin source, d’Ischia et al. have compelled a thorough collection of experimental protocols and recommended procedures for melanin isolation [21].

## 3. Physicochemical Properties

### 3.1. Solubility and Reactivity

Due to melanin’s distinctive solubility and reactivity properties, a first step towards its identification and characterization is usually represented by conventional physicochemical tests. Their role is to endorse, in an early phase, that the isolated pigment could be melanin, so more advanced methods can be further applied. These characteristic properties of melanin include low solubility in distilled water, most organic and inorganic solvents, excepting aqueous alkali, resistance to degradation by concentrated acids, bleaching when subjected to the action of potassium permanganate, potassium dichromate, sodium hypochlorite, hydrogen peroxide or other oxidizing agents, positive reaction for polyphenols (FeCl_3_ test), reaction with sodium dithionite and potassium solution, and reaction with ammoniacal silver nitrate solution [2,8,9,12,13,14,26]. 

### 3.2. Thermal Behavior 

The thermal stability of melanins was tested using thermogravimetric analysis (TGA). TGA thermograms and the corresponding derivative thermogravimetric (DTG) curves were used to evaluate the thermal behavior of analyzed melanins. The results reported so far indicate a high resistance to thermal degradation (Table 2). Usually, the TGA thermograms (Figure 3) showed three steps of thermal degradation of samples [10]. The first endothermal peak was mainly due to the evaporation of weakly and/or strongly bound water. An exothermal peak appeared immediately after that, which was mainly due to the loss of carbon dioxide. The main thermal degradation of melanin occurred at very high temperature and was mainly due to decarboxylation events. It has been also shown that resistance of melanins to thermal degradation depends on their origin. Synthetic melanins are, in general, more resistant to thermal degradation at high temperatures than natural melanins [27,28].

### 3.3. Morphology and Size

#### 3.3.1. Scanning Electron Microscopy (SEM)

The scanning electron microscopy (SEM) is a powerful method for the morphological characterization and particle size distribution of different types of melanin [32,33]. Several sample preparation methods described in literatures are considering the sample’s size, shape, state and the conductive properties [34,35,36]. To become conductive, the melanin samples need to be coated first. For this purpose, a thin layer of gold [37,38,39] or gold/palladium alloy [40] is often used. Depending on the melanin source, the granule morphology and size range between 30–1000 nm and melanin granules are usually amorphous with irregular shape (Table 3).

Several SEM images showing the morphology of sepia melanin, at different magnifications, are presented in Figure 4c–f [41].

**Table 3 ijms-20-03943-t003:** SEM analysis of different types of melanin.

Melanin Source	Granule Morphology and Size	Ref.
**Synthetic**	30−60 nm/40–200 nm/100–150 nm/spherical form	[33,35,42,43]
**Fungal**		
*Auricularia auricula*	20–30 nm/Amorphous fragments with a rough morphology/Chunks of amorphous materials lacking a crystalline structure	[34,44]
*Cryptococcus neoformans/* *Cryptococcus gattii*	Amorphous irregular shape	[42]
*Inonotus hispidus*	89.33 nm (average hydrodynamic size)/Irregular spherical and ellipsoidal structures	[23]
*Mycosphaerella fijiensis*	100–300 nm/Spherical granules	[36]
*Rubrivivax benzoatilyticus*	Spherical, ball-like structures	[45]
*Termitomyces albuminosus*	100–400 nm/Thin amorphous plates comprising large clusters of almost spherical compacted nanogranules	[46]
**Bacterial**		
*Streptomyces glaucescens*	Particles are irregular and have a porous structure/small spheres	[37,47]
*Proteus mirabilis*	Particles are rounded aggregates of spherical bodies (some of the particles have a “doughnut” shape)	[48]
*Pseudomonas sp. (marin)*	Amorphous deposit with no differentiable structures	[49]
*Escherichia coli*	Small granules	[50]
**Others**		
Human hair	Ellipsoidal shape	[42]
Sepia ink	85–165 nm/45–230 nm/70–460 nm/40–160 nm/150 nm/100 nm/100–200 nm/Spherical and quasi-spherical shape particles	[33,41,43,51,52,53,54,55,56,57]
Bovine retinal pigment epithelium (RPE)	Spherical/ovoid with diameters of several hundred nanometers and larger	[43]
Alpaca fiber	400–1000 nm/the particles are compactly arranged inside the matrix cells of the fiber	[57]
*Catharsius molossus* L.	Irregularly shape, tiny layered structures, without smooth surface	[15]

#### 3.3.2. Transmission Electron Microscopy (TEM)

Transmission electron microscopy (TEM) it is one of the most important technologies in cell biology because it allows the examination of the finest cell structures with a high resolution of ~20 nm [32,58]. Typically, melanin samples are chemically fixed, then dehydrated by immersion in an organic solvent, followed by embedding with resins, then ultrathin sections are made (~60–80 nm). The final step is represented by post sectioning staining [17,34,52,58,59,60]. Figure 4a,b) presents the multi-scale assembly hierarchical morphology in the sepia melanin [41]. The most relevant results are presented in Table 4.

As a conclusion, the most appropriate technique to evaluate melanin particles sizes and shapes depends on the sample being imaged and the desired information to be obtained. Both SEM and TEM techniques were able to characterize the melanin sample. Which technique a researcher chooses often depends on the availability, versatility, and costs of the methods. In general, there are more SEM instruments installed worldwide. Additionally, the cost of TEM is considerably higher [64].

#### 3.3.3. Atomic Force Microscopy (AFM)

The atomic force microscope (AFM) is a high-resolution imaging technique that implements 3D topographical data for the measurement of intermolecular forces with atomic-resolution [65]. AFM has important advantages over other microscopic methods because it provides measurements at the nanometer scale [66]. 

The aqueous solutions of eumelanin (from *Sepia officinalis*) were sonicated, centrifugated, lyophilized, and then re-dispersed in ultrapure water. The melanin aqueous suspension was deposited on a disc and the surface was further scanned [35,40,67,68]. 

The granule morphology of synthetic melanin before aggregation is approximatively twice the size of the final melanin particles, due to the free space between the substructures. The light scattering unveiled a random aggregation which indicates where particles attach themselves. In the first step, the aggregates shape changes from sphere to coil, while, in the second step, it returns to a spherical shape (200–400 nm) [35,69,70,71].

The small and spherical sepia ink melanin granules have a 100–200 nm diameter and an average interlayer distance of 0.323–0.35 nm. Three main structural motifs have been described for *Sepia* melanin. The filaments, composed by individual or aggregated particles (4.8 nm was the average filament height and 35 nm was the average diameter), were the first motif. The second described motif was the large particles (150 nm height and 200 nm width), usually organized in aggregated structures. The last motif consists of small individual particles (~ 20 nm dimension). [67,72]. Figure 5 presents the AFM images of a typical *Sepia officinalis* large particles eumelanin aggregate [67].

### 3.4. Elemental Analysis

Melanin is characterized as a heterogenic polymer of phenolic nature. The classification of eu-, pheo- and allo- melanin is broadly accepted and it is based on the chemical composition of the monomer subunit type. Elemental analysis is used together with other methods for structural characterization of melanin pigment. Using this approach, melanin can be easily classified into subtypes either by evaluating the nature of the elements from the pigment and/or determining specific element ratios (C:N:S, C:N or S:N).

Eumelanins, formed mainly of indole type units obtained from L-dopa or l-tyrosine oxidation, are known to lack sulfur in their structure. On the other hand, pheomelanins, derived from cysteinyl conjugates of dopa, contain high amount of sulfur in their molecule. Several fungi are able to produce other types of melanin pigments: DHN-melanins, synthesized via the polyketide pathway, and pyomelanins—soluble melanins produced through hydroxyphenolpyruvate and homogentistic acid [68,73]. Fungal melanins include DHI and DHICA monomer units with 6–9% nitrogen or 1,8-DHN with no nitrogen in its structure [74]. The synthetic melanin was usually used as reference of elemental analysis in differentiation of melanin types extracted from various matrices (Table 5) [16,75]. 

Few remarks are to be taken into account: carbon loss is accentuated at higher pH, acid treatment results in increase of C/N ratio [16], S/N ratio for pure pheomelanin reaches 0.485, while for dopa or dopamine melanin lower ratios were reported (0.005) [75]. C:N and C:N:S molar ratio can give insights into the structure of the oligomer units: eumelanic oligomer units of dihydroxyindole (DHI) have an 8:1 C:N ratio, while 9:1 C:N ratio was associated with dihydroxyindole carboxylic acid (DHICA) oligomer units of eumelanin. Oligomer units of sulfur containing melanins namely benzothiazine and benzothiazole have C:N:S molar ratios of 8:1:1 and 7:1:1 respectively [76]. 

When studying fungal melanin, differentiation between DHN and eumelanin elemental analysis is not sufficient to identify the pigment type. In this case, a study of DHN and Dopa-melanin melanogenesis inhibition using specific inhibitors should be considered [29,70].

**Table 5 ijms-20-03943-t005:** Elemental analysis of melanin.

Precursor/Source of Melanin	Results (%)								Ref.
	C	H	N	O	S	S/N	C/N	C/H	
						Molar Ratio			
Dopa-melanin, 12.5 mM, tyrosinase	51.55	3.79	7.51	-	0.09		8.00		[16]
Dopa-melanin, 12.5 mM pH 8.0	55.25	3.32	8.13	-	-		7.92		
Dopa-melanin, 12.5 mM pH 10.0	47.26	3.28	7.62	-	-		7.24		
Dopamine-melanin, 12.5 mM, tyrosinase	51.25	4.50	7.93	-	0.11		7.54		
DHl-melanin, 12.5 mM, tyrosinase	51.12	3.84	7.68	-	-		7.76		
DHIC-melanin, 12.5 mM, tyrosinase	44.24	4.16	6.16		0.00		8.37		
Dopa-melanin 1 mM	56.45	3.15	8.49		0.09	0.00			[75]
Dopa: 5-S cysteinyldopa 0.9:0.1	55.49	3.62	8.22		1.48	0.07			
0.75:0.25	54.00	3.64	8.45		4.05	0.21			
0.5:0.5	50.96	4.01	8.90		7.17	0.35			
0.25:0.75	48.82	4.24	9.33		9.38	0.44			
5-S cysteinyldopa	48.47	4.08	9.66		10.72	0.48			
Pheomelanin Dopa: L-cysteine 1:1.5	46.24	4.46	9.34		9.78	0.46			
Dopamine melanine	53.78	4.08	7.66		0.11	0.00			
5-S cysteinyldopamine	47.86	4.86	10.85		11.91	0.48			
Synthetic commercially available melanin Precursor: Tyrosine Oxidation by hydrogen peroxide	47.80	3.50	6.15					1.11	[77]
	69.72	-	5.08	24.71	n.r.				[19]
	48.15	3.75	6.77				7.12		[78]
	48	n.r.	7		n.r.				[76]
DHI-melanin	51.52	3.45	7.80						[79]
DHICA-melanin	47.43	3.57	6.42						
Dopamine: cysteine Autooxidation 37 °C, 3 days	50.66	3.65	8.41	3.17				1.14	[80]
Dopamine: cysteine Autooxidation 37 °C, 3 days + freshly dissected putamen tissue, 37 °C for 48 h	51.10	5.12	7.22	3.07				0.83	
MELex5—Precursor: Dopamine	68	5	3		n.r.		26:1		[76]
MEL1—Precursor: Dopamine	48	3	7		n.r.		7.9:1		
MEL2—Precursor: L-DOPA	52	4	7		n.r.		8.4:1		
MEL3b—Precursor: L-Cysteine, L-DOPA, 3:2 molar ratio	34	5	11		22		3.6:1		
MEL4—Precursor: 5-S-cysteinyl-L-DOPA 20:1 molar ratio	46	4	9		10		5.9:1		
n.r. = not reported									

## 4. Spectral Properties

### 4.1. UV-visible Light Absorption Spectrum

The UV-visible absorption spectrum of melanin has been mainly used to identify and characterize the extracted pigment in a preliminary stage, but methods have been also developed for differentiation of melanin types or for quantitative purposes. 

For most types of melanin, the maximum absorption wavelength of alkali solutions ranges between 196–300 nm, depending on the melanin source. Alkaline melanin solutions exhibit a strong optical absorbance in the UV region, which gradually decreases towards longer wavelengths. The high UV light absorption of melanin is apparently due to the complex conjugated molecules in the melanin structure which absorb and scatter photon of UV light [23,47]. The following decrease of absorption is almost linear for most melanins. Therefore, when the logarithm of absorbance of an alkaline melanin solution is plotted against the wavelength, straight lines with negative slopes are obtained. These slopes of linear plots are often used as an important criteria for the identification and characterization of melanins [8,9,23]. In addition, the lack of any absorption peaks at 260 nm and 280 nm suggests the absence of any nucleic acid, lipid or protein impurity [11,22,81]. 

As an example, we recorded the UV spectrum of synthetic melanin and melanin from *Sepia officinalis* (Sigma Aldrich, St. Louis, MO) (Figure 6a). The samples were dissolved in 0.1 M NaOH solution after 5 min sonication and their spectra were recorded using Specord Plus UV/VIS spectrophotometer (Analytik Jena AG) in the range 220–800 nm. All samples were referenced to 0.1 M NaOH. Both melanin samples displayed specific UV spectra, with maximum absorption peaks around 220 nm followed by progressive decrease towards longer wavelengths. After plotting the logarithm of absorbance against the wavelength, negative slopes were recorded for both synthetic (−0.0027) and sepia (−0.0019) melanin (Figure 6b). Similar results were obtained for melanin extracted from various sources, as described in Table 6.

The human epidermal melanin and the neuromelanin extracted from putamen, premotor cortex, *cerebellum*, and *substantia nigra* exhibit a similar broadband absorption spectrum, increasing monotonically towards higher energies [89,90,92]. In the VIS domain, the epidermal melanin has displayed distinctive mechanisms of absorption, mainly due to its distinctive particulate and soluble forms. The absorbance of insoluble melanin increases almost linearly from 800 to 400 nm, while the absorbance of soluble melanin shows an exponential increase at wavelengths shorter than 500 nm. These forms have been further correlated with their molecular weight [90]. The distinctive mechanisms of absorption have been confirmed later, when UVA exposure of pheomelanin solutions showed distinctive spectral and dose–response characteristics from eumelanin, which could help in differentiating these two types of melanin [93]. Moreover, the relative amount of in vivo melanin can be determined using the slope of the absorption spectra from 620–720 nm [89]. 

For human and mouse hair melanin, the relative ratio of eumelanin and pheomelanin can be determined based on the absorbance ratio at 650–500 nm (A_650_/A_550_) after Soluene-350 solubilization. However, the method presents several disadvantages due to the significant background absorbance of residual proteins and the solvent viscosity [4]. 

### 4.2. FTIR Analysis 

One of the most important methods to identify and characterize melanin structure is Fourier-transform infrared spectroscopy (FTIR). Sample preparation for FTIR analysis usually implies grinding the melanin powder with spectrometry grade KBr. The homogenous mixture is then pressed into disks or tablets and scanned using a FTIR spectrophotometer [8,10,81]. 

Although the IR spectra of fungal melanin are slightly different depending on the melanin type and extraction procedure, there are several characteristic bands which can be trailed to identify the major functional groups of the melanin macromolecule. 

Figure 7 presents the comparative FTIR spectra of synthetic and *Sepia officinalis* melanin (both from Sigma Aldrich, St. Louis, MO, USA). The melanin-KBr disks were scanned using a FT-IR Vertex 70 Spectrometer, equipped with Raman RAM II module and IR probe module for non-destructive analysis. The acquisition of FTIR spectra was performed in transmission, with conversion into absorbance and vector normalization, on the spectral domain 400–4000 cm^−1^, spectral resolution of 4 cm^−1^ and 64 scans/sample.

As seen in Figure 7, the FTIR spectra of melanin usually include characteristic bands covering 3600–3000 cm^−1^, 1650–1600 cm^−1^ and 1500–1400 cm^−1^. The strong, broad absorption band observed at 3600–3000 cm^−1^ has been assigned to the stretching vibrations of the -OH and -NH groups belonging to the amine, amide, carboxylic acid, phenolic and aromatic amino functions present in the indolic and pyrrolic systems (6,9,11–13,17). The spectra of various melanins also recorded absorption peaks between 2950–2850 cm^−1^ and between 1465–1375 cm^−1^, which were ascribed to the stretching vibration of aliphatic C-H groups (9,11,16,18,19,21). The strong, characteristic absorption band between 1650–1600 cm^−1^ was usually attributed to the vibrations of aromatic C=C and C=O stretches of carboxylic function [12,23,47]. Specific melanin stretches were observed also between 1500–1400 cm^−1^, which were attributed to the bending vibration of N-H and the stretching vibration of C-N (secondary amine) of an indolic system (9,17,30). Several melanin spectra have also included absorption bands between 1250–1180 cm^−1^, caused by the stretching vibration of phenolic -OH groups [10,47]. The weak absorption bands identified between 1150–1100 cm^−1^ could be a result of the symmetric contraction vibration of C-O-C bond [84]. Below 1100 cm^−1^, the absorption bands were usually weak and were assigned to in-plane deformation of C-H bond, aromatics CH groups or alkene CH substitution/conjugated systems [10,15,23,26]. However, depending on the melanin source, other specific bands have also been identified (Table 7). 

## 5. Nuclear Magnetic Resonance (NMR) Analysis

In order to further confirm the molecular structure of extracted melanin, proton and carbon nuclear magnetic resonance (^1^H-NMR, ^13^C-NMR) analysis can be carried out. First, the sample is dissolved in a deuterated solvent such as deuterium oxide/sodium deuteroxide (D_2_O/NaOD) [11,22] or dimethylsulfoxide-D6 (DMSO-D6) [15,86]. The sample is then subjected to NMR analysis using various resonant frequencies and usually referenced to tetramethylsilane [14,15,86].

There are several chemical shifts in the ^1^H-NMR spectra of melanin which can support the correct identification of melanin molecular structure. As an example of characteristic spectra, the ^1^H-NMR and ^13^C-NMR spectra obtained for the intracellular melanin extracted from *Lachnum singerianum YM296* mycelium is presented in Figure 8 [84].

In the aromatic regions, the chemical shifts between 6–8 ppm were attributed to the aromatic hydrogens of indole and/or pyrrole rings of the melanin polymer chain [22,47,84,85,86]. Between 3.2–4.2 ppm, the observed peaks were caused by the methyl or methylene groups attached to nitrogen and/or oxygen atoms [22,84,85,86]. The presence of a NH-group linked to indole was attributed to the signals within the range of 1.3–2.5 ppm [84,85]. Also, the chemical shifts between 0.2–2 ppm were usually attributed to the stretching vibration of the alkyl fragments (methyl or methylene groups), which indicated traces of residual proteins [15,84,85,86]. 

The ^1^H-NMR spectrum of the commercially available synthetic melanin was recorded in our laboratory (Figure 9). Synthetic melanin (Sigma Aldrich, St. Louis, MO) was dissolved in deuterated 1M NaOH in a concentration of 10 mg mL^−1^ and filtered using 0.22 µm filter. The ^1^H-NMR spectrum was acquired using a Bruker Avance NMR spectrometer operating at 400 MHz proton Larmor frequency (9.4 T magnetic field). 

In the ^13^C-NMR spectra, the peaks between 160–185 ppm were typically ascribed to the carbon atoms in carboxyl and amide groups or the carbonyl groups of peptidic bonds [15,22]. Next, the peaks identified between 120–140 ppm could be generated by aromatic carbons, probably involved in indole or pyrrole systems. The resonance peaks of carbon atoms linked to a nitrogen or sulfur were visible within 50–60 ppm, while the peaks of carbon atoms from methyl and methylene groups were detected within 10–40 ppm (Figure 8b) [84,85]. The ^1^H and ^13^C-NMR resonance peaks observed for several melanin types are listed in Table 8.

## 6. Mass Spectrometry 

Along other techniques, mass spectrometry has been used for the characterization of natural and synthetic melanins, employing different ionization modes, such as electrospray ionization (ESI), electron impact (EI), fast atom bombardment (FAB), matrix assisted laser desorption/ionization (MALDI), secondary ion mass spectrometry (SIMS) or laser desorption synchrotron postionization (synchrotron-LDPI) mass spectrometry. Considering the polymeric form of melanin [81] and the different melanin subtypes, the reported data was rather heterogenous. 

### 6.1. Electrospray Ionization (ESI)

The group of Ye et al. had published several studies [11,85,94,95] involving the structural characterization of non-water-soluble extracellular melanins from different species of *Lachnum* fungi. In all cases, the samples were analyzed by ESI-MS, positive mode, finding that the molecular masses of melanins were significantly different (348 Da—melanin from *Lachnum* YM-346, 432 Da—melanin from *Lachnum* YM-404, 522 Da—melanin from *Lachnum* YM-205, 1172 Da—melanin from *Lachnum singerianum* YM-156). Mass spectra, together with other complementary analysis (NMR, elemental analysis etc) have been used to predict several possible structures.

An extracellular melanin-like pigment produced by *E.Coli* was characterized by Amin et al. [50] by LC-MS/MS, the sample beings dissolved in DMSO. Melanin was found to be present in two forms (m/z 475.5 and m/z 701.5), with the second peak being considered a dimeric form. However, MS/MS analysis was unable to clearly differentiate the possible monomers.

The study of L-DOPA self-aggregation was reported by Li et al. [96]. L-DOPA aggregates were analyzed after just 30 min of aggregation time (in ammonium acetate buffer, 50 mM, pH 8.6). Dimers were found to be predominant, but a wide range of oligomeric species were also identified. It was proposed that L-DOPA, together with 5,6-DHI and 5,6-DHICA, are the building blocks of eumelanin. Even more, considering that no small molecules (e.g., hydrogen or water) were observed to be eliminated during the self-aggregation of L-DOPA, melanin could partially be a noncovalent supramolecular aggregate.

A recent paper by Yacout et al. [97] described ESI-MS experiments conducted on peroxide-treated synthetic melanin. The obtained spectrum had a m/z of 210 as the most abundant ion, suggesting a partial oxidation of DHICA. Still, its structure has not been elucidated, even after MS^3^ fragmentation.

### 6.2. Matrix Assisted Laser Desorption Ionization (MALDI)

MALDI-MS techniques have been widely used for structural characterization of synthetic and natural melanin (Table 9). The aim was to better understand melanogenesis and oligomerization or to characterize melanin extracted from different media. In natural melanin samples, MALDI was used not only to characterize the polymer, but also to identify the melanin type by detecting the monomeric units.

#### 6.2.1. Synthetic Melanins

Over the years, one of the main topics in melanin research was melanogenesis. In most cases, in vitro models were used, where melanin-like pigments have been synthesized through enzymatic oxidation staring from different precursor molecules. In most of the cases, MALDI was the method of choice in characterizing the oligomerization process. Moreover, in most studies, synthetic melanins were used as the reference material for characterizing natural melanins.

The group of Traldi et al. published significant studies on melanogenesis, characterizing different types of melanin biosynthesized starting from tyrosine [98], serotonin [99], dopamine [100], DHICA [101], DHI [102], 5,6-dihydroxytryptamine (5,6-DHT) [103], and DOPA [104] using MALDI. The analysis of melanin biosynthesized from of tyrosine [98] and serotonin [99] revealed that the most abundant ionic species were around 1.7 kDa. Other species were present at 7400 Da, in the 20 kDa and 40 kDa regions, up to 80 kDa, once again underlining the heterogeneity of the melanin polymer and the wide distribution of the molecular masses.

The oligomerization products resulted from melanin biosynthesis from dopamine was investigated in several publications [100,104,105,106,107], highlighting the formation of oligomeric clusters whose intensities are dependent on reaction time. A comparison between melanin biosynthesized starting from DOPA and from dopamine has been carried by Bertazzo et al. [104]. As opposed to dopamine, DOPA seems to be metabolized to DHI, consequently forming DHI melanin oligomers. Later, the same group concluded that UV irradiation has a favorable effect on melanin biosynthesis starting from DOPA by activating multiple reaction pathways [105]. The formation of melanin from dopamine, DOPA, DHI, and DHICA was also studied by Kroesche et al. [106] observing the formation of dopamine oligomers up to 11 units. Even though in the process of polymerization DOPA oligomers are expected to be formed, a study reported a DOPA melanin molecular mass of 2770 Da, with no oligomers detected [107]. The DHICA melanin assessed using the same technique yielded oligomeric species in the range of 500–1500 Da [101]. A more recent paper [108] explored the formation of oligomers over a winder mass range, finding that oligomers of DHI melanin are formed up to 30 units, while N-methyl-5,6-dihydroxyindole (NMDHI) forms melanin oligomers that are up to 45 units, giving specific insights into the synthesis mechanism of eumelanin. By analyzing 5,6-DHT melanin, it could be observed that the oligomerization takes place immediately, forming oligomers up to nonamers [103]. Another form of melanin, the so-called thiomelanin, synthesized starting from the sulfur analog of DHI, 5,6-dihydroxybenzothiophene, was also assessed using MALDI-MS techniques [109].

It can be easily observed that the number of oligomeric species resulted after synthesis is mainly dependent on the nature of the precursor monomer and the reaction conditions. One of the main drawbacks of this approach for melanogenesis characterization is that in all of the cases, the resulting polymer has a homogenous structure, unlike the natural melanin which is known to be formed by several different monomers and to incorporate other molecules (proteins, neurotransmitters, xenobiotics etc. [110]).

#### 6.2.2. Natural Melanins

MALDI-MS proved to be a useful tool in analyzing the melanin pigment in natural samples. The approaches that were used varied from profile comparisons to targeted techniques, as described below.

A recent study, by Yacout et al. [97], pursued the characterization of melanin contained in the retinal pigment epithelium. Considering the polymeric nature of melanin, no exact *m*/*z* can be assigned to it. Still, samples could be differentiated on the *m*/*z* distribution, melanin obtained from young patients showing a characteristic aggregation of ions in the 75,000 and 100,000 *m*/*z* region, while the profile obtained from a patient suffering of age-related macular degeneration is similar with the one of the peroxide treated melanin, suggesting that extensive melanin degradation occurs in the progression of this disease (Figure 10).

The structural analysis of eumelanin from irides was attempted by MALDI-MS [111]. Three homologous series were observed, showing a difference of 191 Da (corresponding to the mass of DHICA), a result that was obtained only after an enzymatic digestion of the melanin sample. In the nondigested samples, only several peak clusters could be observed around 7, 25 and 49 kDa, suggesting the association of melanin with the protein component. The structures of human and cryptococcal eumelanin have been compared [42] in the context of evaluating the protective role of melanin against fungi. The antioxidant capacity of melanin was found to be the highest when regularly linked DHICA units in reduced form at the catechol moiety are present in its structure [112].

Infrared matrix assisted laser desorption electrospray ionization (IR-MALDESI) has been the method of choice for the determination of relative melanin content in human hair samples [113]. Unlike MALDI, IR-MALDESI does not require an organic matrix to promote desorption and ionization of the target analytes. This technique was implemented using a targeted approach, analyzing pyrrole-2,3,5-tricarboxylic acid (PTCA) content as a quantitative marker of eumelanin. 

A marine sponge containing eumelanin was studied as a new potential extraction source [38]. MALDI-MS was performed as part of a more complex characterization process, offering structural information regarding the structural diversity of the extracted biopolymers. Almost every detected peak could be assigned to simple and mixed oligomer structures of DHICA and pyrrole polycarboxylic acids.

MALDI-MS was used by Speiser et al. [114] for the identification of pheomelanin in the shell-eyes of the chiton *Acanthopleura granulata* based on its degradation product 6-(2-amino-2carboxyethyl)-2-carboxy-4-hydroxibenzothiazole (BTCA).

Guo et al. [115] reported that MALDI-MS spectra of melanin extracted from *Streptomyces kathirae* exhibited only low molecular mass components. Similar results have been reported by Banerjee et al. [116] analyzing melanin produced by *Azotobacter chroococcum*.

DHN-melanin produced by the fungal pathogen *Mycosphaerella fijiensis* was studied by linear-MALDI-TOF [36]. The DHN-melanin molecular mass was reported to be around 8000 Da, while the resulted spectrum had a well-defined spacing of 161.8 Da between peaks, which was associated with 1,8-DHN units (Figure 11). This pattern could be explained by the harsh treatment required for the extraction and purification of melanin, considering that the molecular mass of synthetic DHN-melanin was reported to be above 60 kDa [117].

DHN-melanin related metabolites, such as flaviolin and have been identified as a specific marker of DHN-melanin by UV-MALDI-MS in extracts of *P. griseola* f. *mesoamericana* [59] and in those of *Fulvia fulva* [118]. 

The analysis of melanin extracted from *Catharsius molossus* L. [15] revealed a specific graphite-like layered structure. Fragments between 100 and 600 Da have been identified by MALDI-MS, indicating that the advanced structure of this specific melanin may be maintained by non-covalent bonds.

The dark pigment found in the black oat hull was studied by Varga et al. [119] using multiple analysis techniques. The MALDI-TOF spectrum revealed a series of oligomers distanced by 162 Da, concluding that the monomer unit was p-coumaric acid and the pigment is an allomelanin, since these oligomers are formed from phenolic compounds made from phenoloxidase enzymes.

The high complexity of melanin analysis derived from its polymeric nature and low solubility in most solvents has promoted MALDI as a popular approach, offering a way to circumvent those challenges.

However, considering that no separation occurs during the analysis, MALDI can only offer a snapshot of the sample, providing only the molecular ions of the species present in it. Therefore, this method is less suitable for untargeted approaches. 

### 6.3. Other Ionization Techniques

EI and FAB mass spectrometry have been used for the characterization of melanin biosynthesis from dopamine by tyrosinase [120]. Combining these two ionization techniques allowed for the identification of transient intermediates (FAB), but also enabled their structural identification (EI). 

The effect of bleaching on hair melanin has been studied by nanoscale secondary ion mass spectrometry (NanoSIMS) [121], indicating that bleaching increases the oxygen content in melanin granules. 

An attempt to elucidate the avian melanin chemical composition has been made by Yi Liu et al. [122] using Laser Desorption Synchrotron Postionization (synchrotron-LDPI) mass spectrometry. Using this approach, a correlation between feather color and melanin spectral profile could be generated by detecting the specific markers of eumelanin (DHI and DHICA) and pheomelanin (benzothiazole, benzothiazine and isoquinolone).

**Table 9 ijms-20-03943-t009:** Synthetic and natural melanins analysis using MALDI-MS.

Melanin Source	Melanin Type	MALDI Matrix	Scan Range (Da)	Ref.
**Synthetic**	Tyrosine melanin Serotonin melanin	Sinapinic acid or DHB	50–100,000	[98,99]
	Dopamine melanin	DHB	200–3000	[100]
	DHICA melanin	DHB	100–2000	[101]
	DHI melanin	DHB	100–2500	[102]
	5,6-DHT melanin	DHB	100–3500	[103]
	DOPA and dopamine melanin	DHB	100–1200	[104]
	Tyrosine, DOPA and dopamine melanin	DHB	50–2000	[105]
	Dopamine, DOPA, DHI and DHICA melanin	CHCA	n.r.	[106]
	DOPA melanin	CHCA	n.r.	[107]
	DOPA melanin	CHCA	n.r.	[123,124]
	DHI and NMDHI melanin	DHB	500–10,000	[108]
	Thiomelanin Eumelanin	DHB	n.r.	[109]
**Natural**				
Retinal pigment Epithelial melanin	Eumelanin	Sinapinic acid	30,000–175,000	[97]
Hair and iris melanin	Eumelanin	DHB	n.r.	[111]
Human and cryptococcal melanin	Eumelanin	CHCA	380–2000	[42]
Human hair	Eumelanin	No matrix	n.r.	[113]
*Erylus mamillaris*	Eumelanin	n.r.	n.r.	[38]
*Erylus discophorus* var. *deficiens*				
*Pachymatisma johnstonia*				
*Dercitus bucklandi*				
*Acanthopleura granulata*	Pheomelanin	CHCA	n.r.	[114]
*Streptomyces kathirae*	n.r.	DHB	n.r.	[115]
*Azotobacter chroococcum*	n.r.	CHCA	n.r.	[116]
*Mycosphaerella fijiensis*	DHN-melanin	DHB	1000–8000	[36]
*Pseudocercospora griseola* f. *mesoamericana*	DHN-melanin	Norharmane	n.r.	[59]
*Fulvia fulva*	DHI melanin	DHB	n.r.	[118]
*Catharsius molossus* L.		CHCA	n.r.	[15]

5,6-DHT = 5,6-dihydroxytryptamine; CHCA = α-cyano-4-hydroxycinnamic acid; DHB = dihydroxybenzoic acid; DHI = 5,6-dihydroxyindole; DHICA = 5,6-dihydroxyindole-2-carboxylic acid; NMDHI = N-methyl-5,6-dihydroxyindole; n.r. = not recorded.

## 7. Separative Methods Coupled with Mass Spectrometry

### 7.1. Pyrolysis Gas Chromatography Analysis (py-GC-MS)

Due to their polymeric-like structure, melanin pigments have been extensively analyzed by pyrolysis gas chromatography coupled with mass spectrometry detection (py-GC-MS) [125,126,127,128,129]. Briefly, this technique degrades macromolecules by heating them to temperatures high enough to cause bond dissociation. The mechanisms involved are reproducible, the pyrograms obtained being characteristic to the original analyte. Moreover, the pyrolysate retains the original material structure information, permitting differentiation between pure analytes and a mixture or copolymers. This provides a clear advantage when characterizing melanin, especially in the light of recent papers reporting a mixed eu- and pheo- pigment type component of studied melanins [129,130]. 

#### 7.1.1. Synthetic Melanins

To differentiate between different types of natural melanins, the pyrolysis behavior of synthetic melanins was studied and used as a reference (Figure 12, Table 10). 

Some representative studies were made by Dworzański et al. [125,126,127]. Synthetic eu-melanin models obtained from dopa, dopamine and L-tyrosine have been characterized by Py-GC-MS, reporting the effects of oxidation by potassium hexacyanoferrate (III) and reduction with borohydride [125]. Other studies assessed the impact of polymerization conditions (e.g., DOPA concentration, time of oxidation) on the melanin structure [126] and the structural changes of synthetic dopamine and adrenaline melanin induced by copper ions [127]. 

Synthetic neuromelanins were obtained by Dzierżęga-Lęczna et al. [129] from 5-S-cysteinyldopamine, dopamine and polymerization of both precursors (DA/CysDA-melanins) at various molar ratios (9:1, 3:1, 1:1, 1:3). The py-GC-MS experiments conducted emphasized benzothiazine-, benzothiazole- and thiazoloisoquinoline-derivatives as potential biomarkers of 5-S-Cys-Dopamine pheomelanin. Moreover, these compounds were also identified in pyrolisates of Dopamine:Cys-Dopamine-melanin copolymers. 1,2-benzenediol and indole (characteristic for dopamine melanin) were not identified in pyrograms of Dopamine: Cys-Dopamine-melanin. Instead, a correlation between the number and the amount of pyrolytic products originated from 5-S-Cys-Dopamine-derived units and 5-S-Cys-Dopamine content of the copolymer was demonstrated. The same group studied the substantial quantitative changes in pyrolytic patterns of synthetic neuromelanin obtained from dopamine, 5-S-cysteinyldopamine and copolymer dopamine: 5-S-cysteinyldopamine (1:1) after peroxynitrite treatment [130]. The overall conclusion of the study was that peroxynitrite is capable of modifying both eumelanin- and pheomelanin-type monomer units, although the induced structural modifications depend strongly on both melanin type and peroxynitrite concentration.

The Dzierżęga-Lęczna group published an optimized Py-GC-MS with multiple reaction monitoring method for quantitation of pheomelanin in copolymers [133]. In this study, 7-methyl-5H-1,4-benzothiazin-5-one and 7-ethyl-2,3-dihydro-5H-1,4-benzothiazin-5-one were added to the marker panel list for pheomelanin characterization. It was concluded that the type and the relative content of pheomelanin markers depends on whether cysteinyl conjugates of dopa or dopamine are the precursors of the studied pigment. 

Several relevant conclusions can be drawn from data obtained on synthetic melanin. First, Py-GC-MS allows specific and highly sensitive detection of both eu-melanin and pheo-melanin using a panel of markers for each melanin type. Second, the synthesis conditions are critical—the precursor concentration, the polymerization time, the metal ions present and further treatments (e.g reduction and oxidation by various chemicals, acid treatment for isolation of the pigment) have both qualitative and quantitative impacts on the pyrolisate composition. Although synthetic melanin may not be the perfect model for studying natural melanin (due to simplicity of the model—natural melanin is known to be associated with protein, β-linked glucans and metal ions and this association has great impact upon melanin pigment structure), numerous observations can be extrapolated [134].

#### 7.1.2. Natural Melanins

Py-GC-MS has been extensively used in conjunction with other techniques for the structural characterization of melanin pigment isolated from various species (Table 11). For example, Ye et al. studied the melanin extracted from *Lachnum* YM205 mycelia [85]. GC-MS analysis of the pigment showed a high content of pyrrole, benzene and their derivatives. Pigment solubilization was achieved by carboxymethylation and both pigments were tested for in vivo promotion of lead excretion from tissues [85]. 

The extracellular melanin was investigated in several studies [11,135]. Particularly, DeAraujo et al. [135] highlighted the presence of two phenolic compounds, 4-methoxy benzeneacetic acid and 4-methoxy benzenepropanoic acid in the structure of the pyomelanin pigment extracted from *Penicillium chrysogenum*. Using Py-GC-MS, certain aromatic compounds derived from tyrosine degradation were identified. Still, it was not possible to conclusively identify the constituents of the brown pigment, or to confirm if the pigment is pyomelanin. Purification, structure and anti-radiation activity of extracellular melanin extracted from *Lachnum* YM404 were addressed by Ming Ye group [11]. The broth obtained after fermentation was filtered and subjected to the same procedure as reported by Ye et al. [85]. Pyrograms highlighted pyrrole, benzene and their derivatives as major components. Acetic acid, indole, phenol and its derivatives were also detected in the pyrolyzed products. The authors concluded that the studied samples can be classified as indol-pheomelanin pigments [11].

In the study of *Boletus griseus*’s melanin pigment [22], the most abundant products were benzene and indole derivatives, the characteristic thermal degradation products of eumelanin [136]. Also, few quinoline and isoquinoline molecules were detected, confirming a mixed eumelanin-pheomelanin structure of the analyzed pigment. 

**Table 11 ijms-20-03943-t011:** py-GC-MS analysis of natural melanins.

Melanin Source	Py-GC-MS Results	Ref.
**Fungal**		
*Lachnum YM205* mycelium	Main pyrolysis products: pyrrole, benzene and their derivatives	[85]
*Lachnum YM404* extracellular melanin	Main pyrolysis products: pyrrole, benzene, and their derivatives; 2-methylpyrrole, 3-methylpyrrole and 2, 3-dimethylpyrrole; toluene, ethyl- benzene, 1,4-xylene and styrene Acetic acid, indole, phenol and its derivatives were also detected	[11]
*Lachnum singerianum YM296* mycelium	Main pyrolysis products: benzene, pyrroles, phenols, indoles, benzonitriles, carboxylic acids and sulfocompounds; phenol, 1,2- benzenediol, benzonitriles and carboxylic acids (ethanoic acid and phthalic acid) small amount of sulfur compounds (thiazole and benzothiophene)	[84]
*Boletus griseus*	Main pyrolysis products: benzene and its derivatives, followed by indole and its derivatives; few quinoline and isoquinoline molecules small amount of phenyl nitrile, furan, and pyrazole	[22]
*Penicillium chrysogenum* cell-free fungal growth medium	Pyomelanin 4-methoxy benzeneacetic acid, 4-methoxy benzenepropanoic acid and other phenolic compounds	[135]
*Drosophila melanogaster* Flies	D. melanogaster type egl and w: methylbenzene, phenol, 2-methylindole, 5-hydroxyindole D. melanogaster yw: methanethiol, benzomethanothiol 2-propyl-1,3-dithiolane, 3-hydroxybenzothiazine	[137]
*Catharsius molossus*	Main pyrolysis products: pyrrole, indole, phenol and their alkyl derivatives; traces of alkyl derivatives of thiophene; higher content of indole and its derivatives; 5,6-diacetoxy-1- methyl indole; hydroquinone and derivatives, dioxoindoline	[15]
*Sepia officinalis*	Main pyrolysis products: pyrrole, indole, phenol and their alkyl derivatives pyridine and its derivatives; 5,6-diacetoxy-1- methyl indole; 5,6-dipropionyl-1-methyl indole	[15]
Epidermal Human Melanocytes	The major pyrolysis product: styrene; pyrrole and its methyl derivatives, toluene, phenols, and indoles Thiazole and its methyl derivative; hydrogen sulfide, carbonyl sulfide, and methanethiol with low retention times TMAH thermochemolysis: N,N-dimethylated amino acid glycine and amine derivatives	[132]
Primary human epidermal melanocytes derived from lightly-and darkly-pigmented neonatal foreskin	The most abundant pheomelanin markers: thiazole and hydroxybenzothiazole Non-sulfur containing pyrolysis products: toluene, phenol, methylphenol, indole and methylindole	[138]
Human melanoma cells (A-375) Treatments: Dimethylsolfoxid (DMSO) Valproic acid (VA) 5,7-dimethoxycoumarin (DMC) Combination of VA+DMC	Main pyrolysis products: pyrrole, indole, phenol and their alkyl derivatives dominated by 1,2- benzenediol *DMSO melanin:* lower content of compounds (pyrrole, indole, and benzene derivatives)	[139]
	*DMC melanin*: Styrene and α-methylstyrene; Toluene, methylethylbenzene and small amounts of benzene, pyridine, pyrrole, phenol, indole and their methyl derivatives *VA melanin*: Benzene, pyridine, pyrrole, toluene, styrene, 4-methylphenol and indole; Benzothiazole; *VA+DMC melanin*: Thiophene, thiazole, pyrrole and its alkylic derivatives, methyl derivatives of pyridine, styrene and α-methylstyrene	[140]
Black human hair	Low yields of indole in pyrolisates; high content of alkilindole derivatives	[125]
Cattle eye melanosomes isolated from retinal pigment epithelium, ciliary body, iris	Main pyrolysis products: Pyrrole and its derivatives phenylacetonitrile, phenols, indoles, catechol and its derivatives	[141]
*Substantia nigra* Brain tissues of neurologically normal adult individuals	High levels of low molecular weight gases with low retention times The most abundant pyrolysis product: limonene; pyrrole, benzene, phenol and indole derivatives Enhanced relative content of pyrrole and indole derivatives in the pyrolysate of proteinase-untreated neuromelanin samples No pheomelanin markers	[131]
	Methylpyrrole, toluene, ethylbenzene, styrene, indole methylindole, Dodecene, tetradceanoic acid pentadecanoic acid and esade- canoic acid High amount of indole derivatives compared to phenols	[142]

Melanin from three different color strains of *Drosophila melanogaster* was isolated and the products formed during thermal degradation of the biopolymers were analyzed by py-GC-MS, having dopa-derived melanin as synthetic model [137]. Several conclusions were drawn: i. benzene, pyridine, pyrrole, phenol, benzenediol, indole, benzenenitrile derivatives, and sulfur containing compounds were the major products identified; ii. the synthetic DOPA-melanin presented a large quantity of alkyl derivatives of pyrrole and phenol derivatives, opposed to the natural melanins from *D. melanogaster*, which contained more benzene, pyridine and indole derivatives; iii. low levels of D-pyrrole derivatives, decreased quantity of H-benzenenitrile derivatives and slightly decreased level of F-benzenediol derivatives were observed in all types of melanin from *D. melanogaster* strains; iv. in terms of pyrolytic profile, the black and grey strains were comparable to DOPA-melanin with regard to the products detected, while the pyrogram of the melanin extracted from the yellow strain was characterized by sulfur-containing products of the pyrolysis. 

Py-GC-MS analysis of melanin extracted from *Catharsius molossus* was conducted by Xin et al. [15], with the results being compared with the commercially available *Sepia officinalis* melanin. Compounds characteristic to eumelanin pyrolysis products were shown in both melanic pigments. In addition, thiophene and derivatives and a higher content of indole were present in the melanin extracted from *C. molussus*, indicating evident traces of pheomelanin. 

The melanin extracted from cultured human melanocytes was structurally studied by py-GC-MS by Stępień et al. The main identified pyrolysis products were styrene, pyrrole and its methyl derivatives. Pheomelanin markers were also detected, but in lower amount: thiazole and its methyl derivative or sulfur-containing low molecular weight gases. Recently, the same group published a study of pheomelanin content from cultured human melanocytes from lightly and darkly pigmented skin using a Py-GC-MS/MS approach [138]. The pyrograms of melanins showed substantial differences in the pheomelanin contents, irrespective of the selected isolation procedure. It was concluded that darkly pigmented neonatal skin contains approximately four times more pheomelanin than the light skin. Py-GC-MS was also used in the study of impact of certain molecules upon melanin structure [139,140]. The pyrolytic profile of melanin derived from human melanoma cells (A-375) treated with DMSO was qualitatively similar to the synthetic melanin obtained from tyrosine by enzymatic procedure, but 1,2- benzenediol was the main pyrolysis compound. Both quantitative and qualitative structural differences were observed in melanin isolated from A-375 cells treated with valproic acid and 5,7-dimethoxycoumarin. Markers for pheomelanin were present in both melanin isolated from valproic acid and a mixture of valproic acid and dimethoxycoumarin treated melanoma cells. Toluene, methylethylbenzene and small amounts of benzene, pyridine, pyrrole, phenol, indole and their methyl derivatives were identified in the pyrograms of all melanin samples studied, indicating the presence of the eumelanin component.

Dworzański’s group published several research articles on melanin pigment extracted from cattle eye, iris, and hair [125,126,141], being the first group to use py-GC-MS to differentiate the pigment types. Melanin extracted from black human hair, bovine eyes and also model polymers derived from tyrosine, 3,4- dihydroxyphenylalanine and 3,4- dihydroxyphenethylamine showed qualitative similarities, although differences in abundance of pyrole, indole, methylcatechol and alkilindole formation were observed [125]. Satisfactory differentiation was obtained as pyrogram plots showed good separation of the various melanin types.

Further, the group investigated the melanin extracted from retinal pigment epithelium, ciliary body and iris of cattle eye. The main pyrolytic products were identified as pyrrole and its derivatives, phenylacetonitrile, phenols indoles and catechols which may be considered to be breakdown products of proteins and melanins. The overall observations were that melanin pigments are highly influenced by the protein composition, acid hydrolysis is able to modify the melanin structure, and further isolation strategies of melanin should focus on milder procedures. 

Neuromelanin’s structure and its physiological function have been intensively studied considering its supposed implication in cell death associated with Parkinson’s disease [131,142]. The pyrolytic patterns of the studied neuromelanins were very similar qualitatively, regardless of the proteinase treatment, with the most abundant pyrolysis product being limonene [136]. Other peaks corresponded to pyrrole, benzene, phenol and indole derivatives, but no sulfur-containing markers of pheomelanin were identified. These results come to reinforce previous conclusions that natural melanin pigment differs structurally from both eumelanin and pheomelanin standards. Also, pyrrole and indole derivatives might be an indicator of peptide moiety bounded to melanin. 

### 7.2. Liquid Chromatography Analysis (LC-MS)

Liquid chromatography can serve as a quantification method for eumelanin and pheomelanin by measuring their degradation products, as first described by Ito et al. [75,143]. The oxidation of eumelanin and pheomelanin yields specific markers, such as PTCA and pyrrole-3,5-dicarboxylic acid for eumelanin, while the hydrolysis of pheomelanin produces thiazole-2,4,5-tricarboxilic acid (TTCA), thiazole-4,5-dicarboxylic acid (TDCA), 4-amino-3-hydroxyphenylalanine (AHP) and 3-amino-tyrosine (3-AT). These molecules were used to quantitatively assess the melanin in various samples, such as hair [75,143,144,145,146,147,148,149], skin tissue [75,143,146,147], fungi [17], irides [147], melanocytes [150], urine [151], fossils [79], and sepia ink [152]. 

For obtaining these specific markers, some sample preparation techniques have been employed. To obtain PTCA, eumelanin was usually oxidized using KMnO_4_, while the pheomelanin specific hydrolysis products were obtained by acidic hydrolysis using hydroiodic acid. Later, a more efficient and less labor-intensive oxidation procedure was proposed [145,146,153] using alkaline H_2_O_2_ oxidation, which is applicable to both eumelanin and pheomelanin containing samples. In this case, for ESI-MS applications, ammonium hydroxide can be used for alkalization [146], even though NaOH and K_2_CO_3_ are typically used.

All but one of the developed methods for the analysis of the melanin hydrolysis products used reversed phase liquid chromatography, using nonpolar C8 and C18 stationary phases and methanol or acetonitrile as organic modifiers. The aqueous phase was usually acidic (acetic acid, fluoroacetic acid, trifluoroacetic acid), while in few cases [75,143,148,149] sodium octanesulfonate was used as an ion pairing agent, for a better retention of the AHP. As an alternative, Nezirević et al. [151] separated 3-AHP and 4-AHP by HILIC. 

Recently, Petzel-Witt et al. [144] described a LC-MS/MS method for the assessment of PTCA in hair samples. The mass spectrometer was operated in multiple reaction monitoring (MRM) mode, employing negative ionization mode, two transitions of PTCA fragmentation being monitored: *m*/*z* 198 -> 154 (quantifier) and *m*/*z* 198 -> 110 (qualifier); the same fragmentation pattern was also reported in another study [146]. 

The analysis of homogentisic acid (HGA) has been used for the characterization of pyomelanin containing samples, considering its role as a precursor of tyrosine catabolism. By analyzing HGA in a cell-free culture broth, Vasanthakumar et al. [135] identified the production of pyomelanin by *P. crysogenum*, while Singh et al. [25] by three species of marine bacteria. The homogentisate pathway has been characterized by Wang et al. [154] in *Aeromonas* media samples, finding that, in this case, pyomelanin biosynthesis starts from phenylalanine, rather than tyrosine. This is contrary to the initial findings [155], which stated that *Aeromonas* media produces a DOPA-derived pheomelanin. 

## 8. Conclusions 

In this review, we presented a comparative evaluation of the analytical methodologies applicable to qualitative or quantitative assessment of melanins from very different matrices, from extraction and purification to high-throughput methods such as py-GC-MS or MALDI-MS. Each time, special attention was given to the particular factors which can highly influence the analysis outcome such as extraction, purification, concentration, etc. 

In conclusion, the analytical tools available for melanin analysis are mostly complementary, offering specific information needed to draw accurate conclusions regarding the sample to be analyzed. If only qualitative information is needed, then the best analysis methods would be those which can classify melanin pigments into its subtypes, such as elemental analysis, py-GC-MS, LC-MS and MALDI. For a quantitative approach, the most desirable workflow would include oxidation or hydrolysis of the sample followed by the LC-MS determination of its degradation products. In the case of structural elucidation, the best approach would be the microscopic techniques (TEM, SEM, AFM), together with MALDI, FTIR and NMR. Also, one should not overlook the basic determinations such as solubility and UV spectra, which can offer valuable information in the shortest time.

The final purpose of all these investigations is to guarantee that melanins are properly characterized to be use in a wide range of biomedical and technological applications, from dermatocosmetics, (especially photoprotection), to radioprotection, thermoregulation, or plant virulence. 

## Figures and Tables

**Figure 1 ijms-20-03943-f001:**
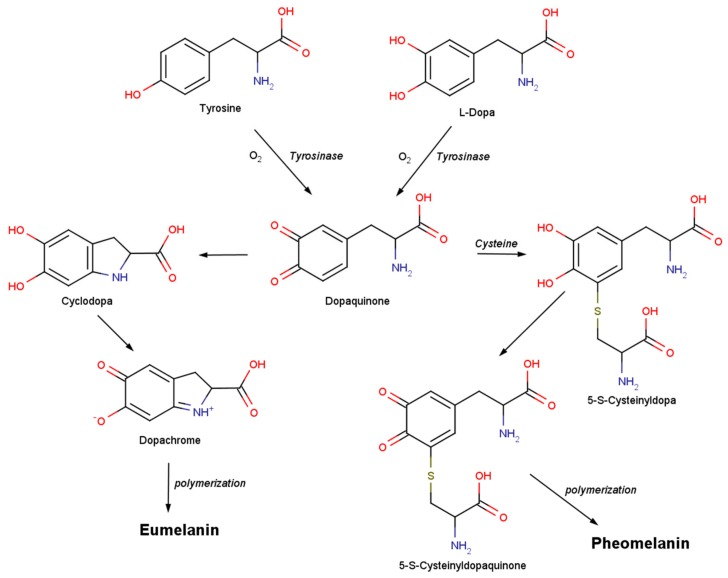
Scheme of eumelanin and pheomelanin synthesis.

**Figure 2 ijms-20-03943-f002:**
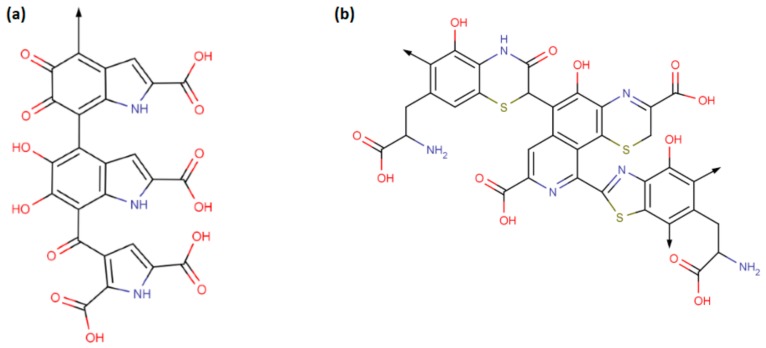
The chemical structures of (**a**) eumelanin and (**b**) pheomelanin as presented in Ref. [4].

**Figure 3 ijms-20-03943-f003:**
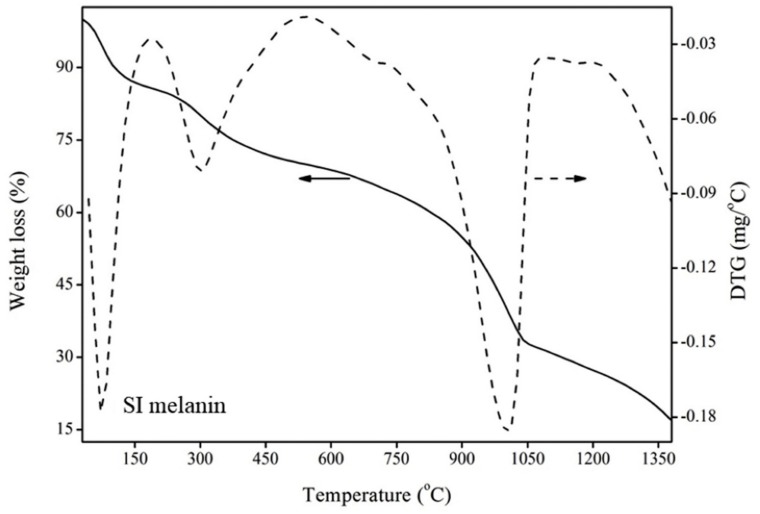
TGA (solid line) and DTG (dotted line) thermogram of melanin isolated from sepia ink (SI melanin) (Partially reproduced with the permission from Ref. [10]).

**Figure 4 ijms-20-03943-f004:**
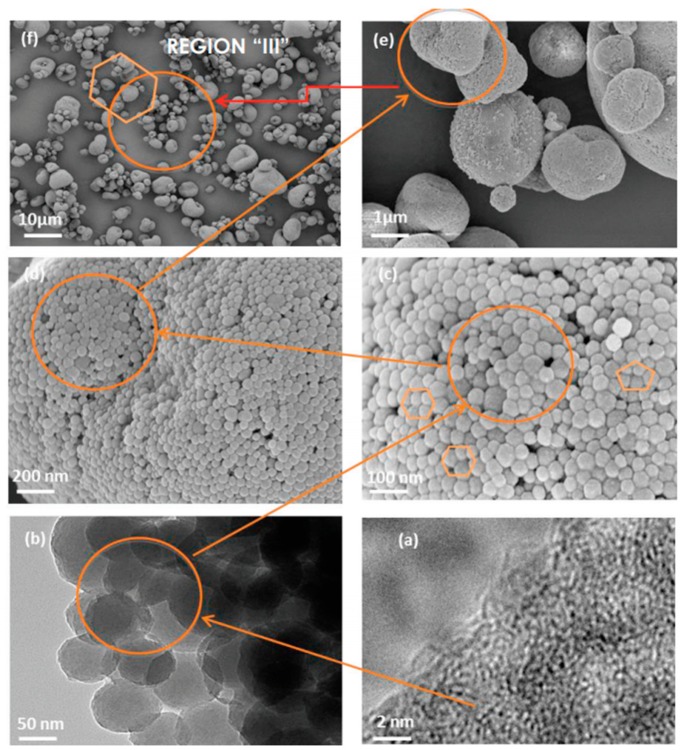
SEM and TEM images showing the multi-scale assembly hierarchical morphology of *Sepia* melanin at different magnifications; melanin nanoparticle scaled at 2nm (**a**); melanin nanoparticle highlighting the internal organization in a form of a block-copolymer-like structure (**b**); secondary self-aggregation of melanin nanospheres in polydisperse large clusters (**c**–**e**); deflated clusters with a wide size distribution form of self-aggregates (**f**) (Reprinted with permission from Ref. [41]).

**Figure 5 ijms-20-03943-f005:**
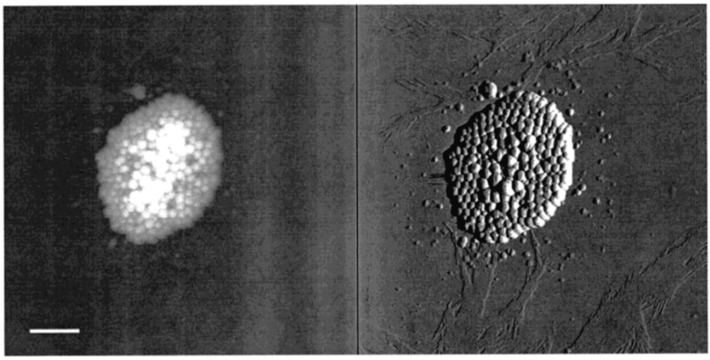
AFM height (left) and deflection (right) images of a typical *Sepia officinalis* eumelanin aggregate (the scale bar is 970 nm) (Reprinted (adapted) with permission from Ref. [67]). Copyright 2001 American Chemical Society.

**Figure 6 ijms-20-03943-f006:**
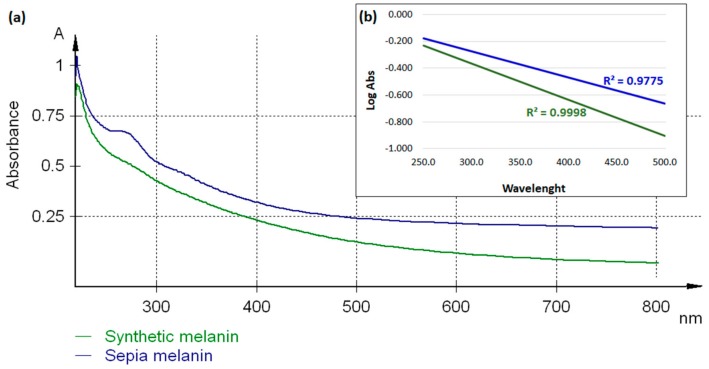
(**a**) The UV-VIS spectra of synthetic melanin (green) and melanin from *Sepia officinalis* (blue); (**b**) The linear curves obtained after plotting the logarithm of absorbance against the wavelength.

**Figure 7 ijms-20-03943-f007:**
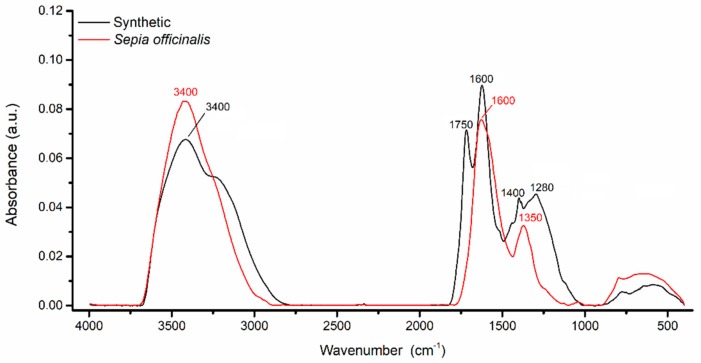
The FTIR spectra of *Sepia officinalis* (red) and synthetic (black) melanin.

**Figure 8 ijms-20-03943-f008:**
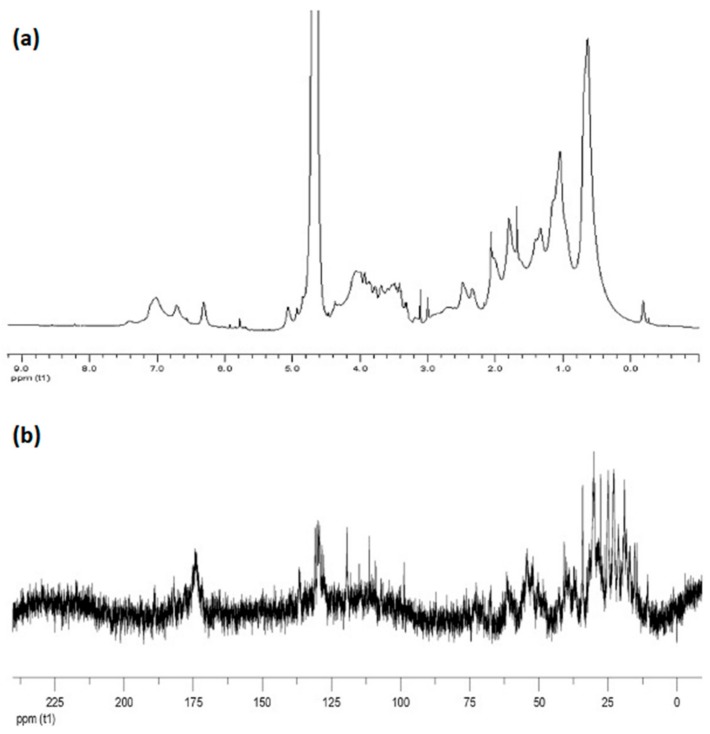
(**a**) The ^1^H-NMR spectrum and (**b**) the ^13^C-NMR spectrum of melanin extracted from *Lachnum singerianum* YM296 mycelium (Partially reproduced with permission from Ref. [84]).

**Figure 9 ijms-20-03943-f009:**
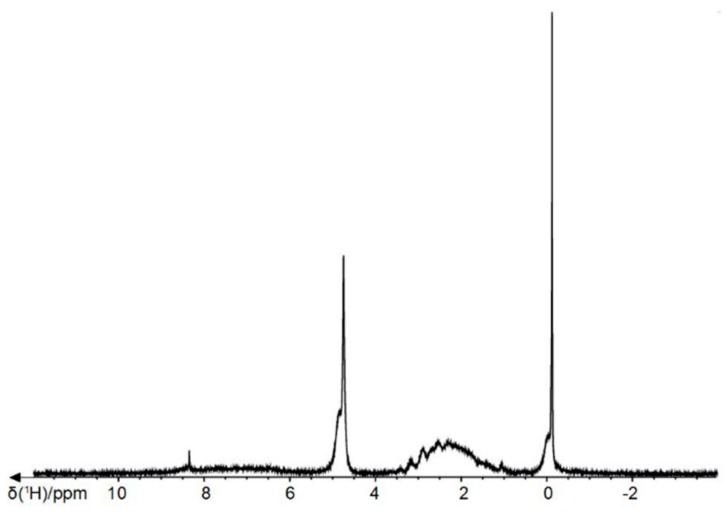
The ^1^H-NMR spectrum of synthetic melanin.

**Figure 10 ijms-20-03943-f010:**
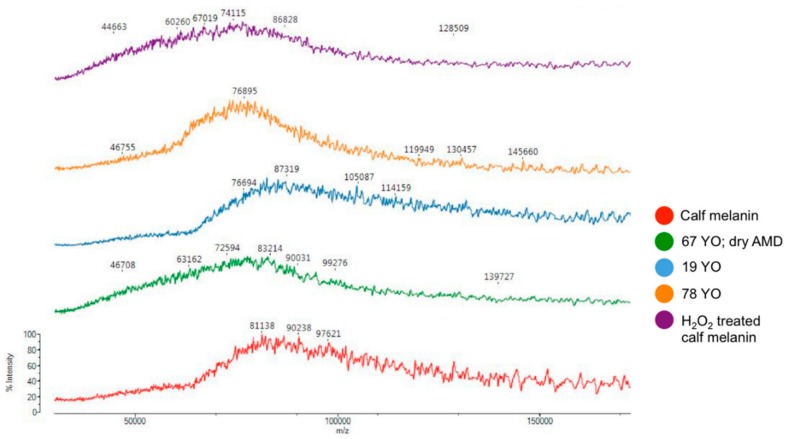
Comparative MALDI-MS profiles of melanin, reproduced with permission from Ref. [97]. (67 YO; dry AMD = 67-year-old donor; dry age-related macular degeneration sample; 19 YO = 19-year-old donor sample; 78-year-old donor sample).

**Figure 11 ijms-20-03943-f011:**
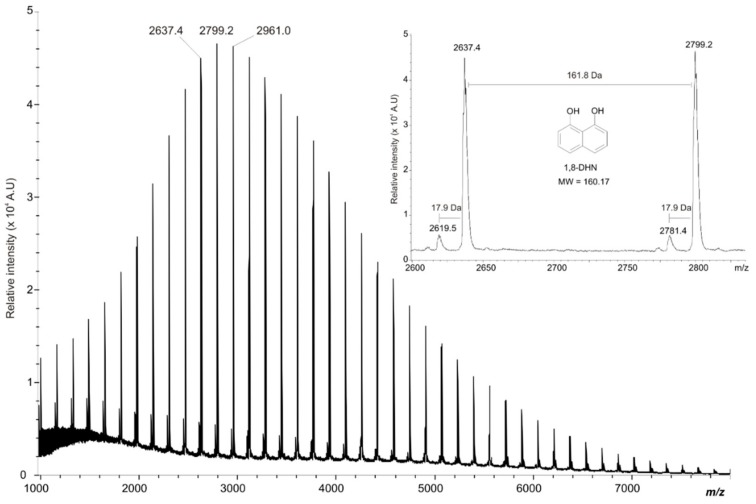
MALDI-TOF of DHN-melanin extracted from *M. fijiensis* mycelium. (Reproduced with permission from Ref. [36]).

**Figure 12 ijms-20-03943-f012:**
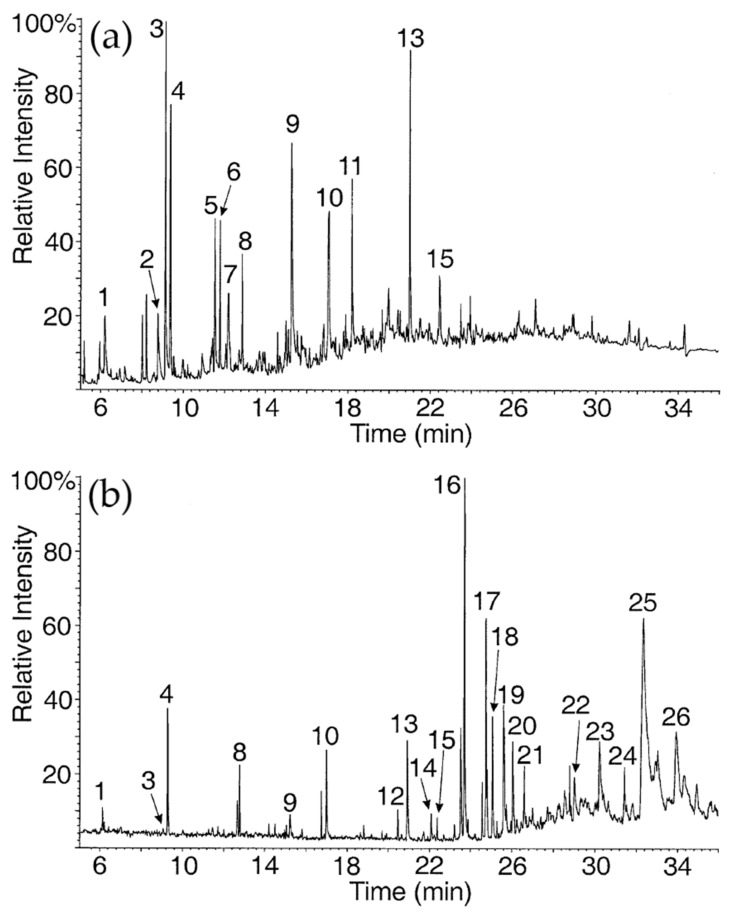
Total ion chromatograms of products formed during pyrolysis of (**a**) synthetic eumelanin standard (DA-melanin) and (**b**) synthetic pheomelanin standard (CDA- melanin). Peak designation: (1) benzene, (2) pyridine, (3) 1H- pyrrole, (4) methylbenzene, (5) and (6) 2-methylpyrrole or 3-meth- ylpyrrole, (7) methylpyridine, (8) ethenylbenzene, (9) phenol, (10) 4-methylphenol, (11) benzyl nitrile, (12) isoquinoline, (13) indole, (14) 4-hydroxybenzothiazole, (15) methylindole, (16) 2H-1,4-ben- zothiazin-5-one, (17) 7-methyl-2H-1,4-benzothiazin-5-one, (18) 4-hydroxy-6-ethyl-benzothiazole, (19)-(21) not identified, (22) and (23) thiazolo [4,5-f]isoquinoline or thiazolo[5,4-f]isoquinoline, (24) 2-methyl-thiazoloisoquinoline, (25) 1H[1]benzothiopyrano[3,4-d]imidazol-4-one, (26) not identified. Reproduced by permission from Springer Nature, Journal of the American Society for Mass Spectrometry, GC/MS analysis of thermally degraded neuromelanin from the human substantia nigra. Dzierzega-Lecznar A, Kurkiewicz S, Stepien K, Chodurek E, Wilczok T, Arzberger T, Riederer P, Gerlach M. License no. 4643461452547; 2004 Jun 1;15(6):920-6. [131].

**Table 1 ijms-20-03943-t001:** Examples of extraction and purification procedures applied to various melanin types.

Melanin Source	Extraction and Purification	Ref.
**Fungal Melanin**		
*Boletus griseus*	• incubation with 1 M NaOH at 60 °C for 40 min • centrifugation followed by pH adjustment to 1.5 with 6 M HCl and heating at 80 °C for 12 h • centrifugation followed by precipitate collection, washing with deionized water • vacuum freeze-drying of the precipitate • hydrolyzation of the crude melanin with 7 M HCl at 100° for 4 h • filtration and washing • repeated solubilization of the precipitate in 1 mM KOH and addition of chloroform: isoamyl alcohol • centrifugation followed by pH adjustment to 2 with 1 M HCl • centrifugation followed by successive washing steps with ethanol and ultrapure water • vacuum freeze-drying of the precipitate	[22]
*Inonotus hispidus*	• Similar extraction and purification steps	[23]
*Auricularia auricula*		[14]
*Lachnum* YM404		[11]
*Pleurotus cystidiosus*		[9]
*P. capitalensis*		[8]
*C. neoformans* *C. sphaerospermum* *W. dermatitidis 8656*	• suspension of fungal cells in 1 M sorbitol-0.1 M sodium citrate (pH 5.5) • lysis using *Trichoderma harzarium* enzymes followed by overnight incubation at 30 °C • centrifugation followed by protoplast collection and denaturation using 4 M guanidine thiocyanate overnight, at room temperature • centrifugation and overnight treatment with Proteinase K, at 37 °C • particle boiling in 6 M HCl for 1 h • wash of the resulting material with phosphate buffer saline and deionized water • dry in air at 65 °C overnight	[17]
*Aspergillus fumigatus*	• Similar extraction and purification steps	[18]
**Sepia ink**	• washing with distilled water • dilution of sepia ink paste with distilled water (1:5) using a homogenizer • centrifugation followed by repeated washing of the melanin precipitate • drying at 40 °C for 24h to obtain dried melanin nanoparticles	[10]
**Human hair**	• repeated hair wash with acetone, dichloromethane, ether, ultrapure water • addition of 0.1 M phosphate buffer and dithiothreitol (DTT); continuously stir the solution for 23 h at 37 °C under argon • repeated overnight incubation with proteinase K and/or papain and DTT, centrifugation followed by pellet wash and re-suspension in phosphate buffer • Triton X-100 treatment followed by 4 h stirring, ultracentrifugation and exhaustive pellet wash with water and methanol • final overnight treatment with proteinase K and DTT; successive washing steps with water and dry over NaOH under argon	[20]
**Neuromelanin**	• brain tissue grinding; addition of water and shaking • centrifugation followed by washing with phosphate buffer • incubation with Tris buffer (50 mM, pH 7.4) containing SDS for 3 h at 37 °C • centrifugation followed by pellet incubation with the same solution containing proteinase K for 3 h at 37 °C • centrifugation followed by pigment wash with NaCl 0.9% and water • suspension of pigment in methanol, centrifugation; resuspension of pigment in hexane, centrifugation • pigment dry under nitrogen flow and placed under vacuum for 14 h at room temperature	[24]
**Bacterial melanin**	• centrifugation of a 6-day culture followed by acidification of the cell-free supernatant with 1 N HCl • storage at 25 °C in the dark for 1 week • 1 h boiling followed by cooling and centrifugation • pellet wash with 0.1 N HCl and double distilled water • pellet wash with ethanol at 100 °C for 10 min • 24 h storage at room temperature • residue wash with ethanol followed by air dry	[25]

**Table 2 ijms-20-03943-t002:** Characteristic peaks on TGA/DTG curves of melanins.

Melanin Source	Endothermal Peak (°C)	Exothermal Peak (°C)	Decomposition (°C)	Ref.
Synthetic				
DOPA-melanin	76.85	424.85	-	[27,28]
DMSO-thin film melanin	150	-	1000	[29]
Natural				
Black garlic	71.5	306.7	915.3	[10]
Sepia ink	71.5	306.7	998.3	[10]
Banana hard core peel	59.85	399.85	-	[27,28]
Bovine eyes	67.85	379.85	-	[27,28]
*Bacillus subtilis*	280	390	500	[30]
*Klebsiella sp. GSK*	220	350	700	[31]

**Table 4 ijms-20-03943-t004:** TEM analysis of different types of melanin.

Melanin Source	Granule Morphology and Size	Ref.
**Synthetic**	Spherical particles with 100–150 nm size	[61]
*Sepia officinalis*	Small spherical granules with 100–200 nm diameter and average interlayer distance of 0.323 nm–0.35 nm. Identified configurations: polymeric fibrils-like; cross-linked fibrils-like; onion-like planar polymeric chains	[32,41,62]
**Fungal**		
*Auricularia auricula*	The internal structure: circular units with a dense cell wall with several concentric layers encapsulated in a heterogeneous mass	[34]
*Cryptococcus neoformans*	Melanin spheres that appear as empty particles, named melanin “ghosts”	[34]
*Wangiella dermatitidis* (DHN-melanin)	Granular structure on the external cell wall structure	[62]
**Human hair**	Largely or partially decomposed	[63]

**Table 6 ijms-20-03943-t006:** The UV-VIS spectra characteristics for different types of melanin.

Melanin Source	Maximum Absorption Wavelength (nm)	Slope *	Ref.
**Fungal**			
*Auricularia auricula*	215	−0.0030	[14,82]
*Boletus griseus*	214		[22]
*Chroogomphus rutilus*	212		[13]
*Exophiala pisciphila*	216	−0.0030	[83]
*Inonotus hispidus*	212	−0.0031	[23]
*Lachnum singerianum* YM296	215		[84]
*Lachnum* YM205	196	−0.0015 to −0.0030	[85]
*Lachnum* YM226	223	−0.0030	[86]
*Lachnum* YM404	210	−0.0015 to −0.0030	[11]
*Mycosphaerella fijiensis*	200–250		[36]
*Ophiocordyceps sinensis*	220–240	−0.0019	[12]
*Phyllosticta capitalensis*	240	−0.0015	[8]
*Pleurotus cystidiosus*	280		[9]
**Bacterial**			
*Streptomyces glaucescens*	250		[47]
**Marine species**			
*Actinoalloteichus sp. MA-32*	300	−0.2646	[87]
*Patinopecten yessoensis*	297		[88]
**Human epidermal melanin**	broadband absorption spectrum		[89,90]
**Others**			
Black garlic	210–250		[10]
Sepia ink	210–250		[10]
Industrially polluted metagenomic library equipped *Escherichia coli*	290		[50]
*Castanea mollissima*	270–280	–0.0050	[91]

* the slope obtained after plotting the log absorbance of the pigment against the wavelength.

**Table 7 ijms-20-03943-t007:** The FTIR absorption bands reported for different melanins.

Melanin Source	Absorption Peaks/Regions (cm^−1^)	Ref.
**Synthetic melanin**	3314; 3107; 1713; 1599; 1217	[42]
	3406; 2924; 1720; 1620; 1296	[82]
**Fungal melanin**		
*Auricularia auricula*	3300–3500; 3399; 2925; 1633; 1075	[14]
	3422; 2923; 2853; 1627	[82]
*Boletus griseus*	3426; 1600; 1384; 1105; 873–618	[22]
*Chroogomphus rutilus*	3400–3200; 2923; 1614; 1422–1344; 1280; 1241	[13]
*Cryptococcus neoformans/* *C. gattii*	3300–3250; 2950–2900; 1620–1630; 1545–1525; 1400–1350; 1050–1000	[42]
*Exophiala pisciphila*	3360–3000; 2924; 2855; 1709; 1618; 1239; 721	[83]
*Inonotus hispidus*	3298; 2934; 1624; 1536; 1402; 800–600	[23]
*Lachnum singerianum* YM296	3388; 2927; 2854; 1629; 1517; 1408; 1115; 620	[84]
*Lachnum* YM205	3280; 2850; 1630; 1540; 1460; 1400; 1240; 1160; 700–600	[85]
*Lachnum* YM226	3130; 1640; 1400; 619;	[86]
*Lachnum* YM404	3156; 3047; 1568; 1663; 1403; 1103; 900–650; 700–600	[11]
*Mycosphaerella fijiensis*	3700–3000; 3433; 2953; 2853; 1711; 1628; 1244; 1026	[36]
*Ophiocordyceps sinensis*	3421; 2929; 1706; 1618	[12]
*Phyllosticta capitalensis*	3352; 1639	[8]
*Pleurotus cystidiosus*	3445; 2925; 1637, 1025	[9]
**Bacterial melanin**		
*Streptomyces glaucescens*	3421; 2947; 1647; 1539; 1423; 1240; 1058; 864	[47]
**Marine species**		
*Actinoalloteichus sp. MA-32*	3346; 2943; 2835; 1446; 1112, 1029	[87]
**Others**		
Human black hair	3277; 2967; 1625; 1522; 1043; 877	[42]
Black garlic	3170; 2922; 1704; 1620; 1514; 1363; 1162; 1021; 796; 502	[10]
Sepia ink	3278; 2920; 2851; 1708; 1645; 1518; 1453; 1411; 1208; 1036; 925; 603	[10]
Industrially polluted metagenomic library equipped *E. Coli*	3424; 2926; 2856; 1640; 1538; 1454; 1232; 750	[50]
*Catharsius molossus* L.	3400; 2950–2850; 1650–1600; 1453; 1400–1380; 800–600	[15]

**Table 8 ijms-20-03943-t008:** The ^1^H and ^13^C-NMR chemical shifts reported for different melanins.

Melanin Source	δ_H_ (ppm)	δ_C_ (ppm)	Ref.
**Fungal**			
*Auricularia auricula*	8.29; 7.02; 6.73; 4.7–5.4; 3.5–4.5; 0.5–2.5;	40–15	[14]
*Boletus griseus*	8–8.5; 6.5–7.5; 5.5–6.5; 4.2–5.4; 3.2–4.2; 0–2.5	175–165	[22]
*Lachnum singerianum YM296*	6.2–7.5; 3.3–4.5; 1.5–2.5; 0.8–1.2	10–40; 50–70; 110–135; 170–185	[84]
*Lachnum YM205*	7.1; 4.2–5.4; 3.1–4.2; 1.7–2.5; 0.2–2	10–40; 50–60; 120–138; 138–163; 170–180	[85]
*Lachnum YM226*	10.74; 9.15; 8.0–7.6; 7.2; 6.9; 6.6; 4.7–5.4; 4.47; 3.9; 3–0.7; 2.5;	175; 195	[86]
*Lachnum YM404*	3.2–4.2; 1.3–2.5; 0.5–1.2	10–45; 55–60; 150–160; 180–185	[11]
**Bacterial**			
*Streptomyces glaucescens*	7.6; 7.35; 7.00; 6.60; 1.0–3.2		[47]
**Others**			
Industrially polluted metagenomic library equipped *E. Coli*	9.2; 7.2–8.0; 6.6–7.1;	171; 100; 120–140; 20–24	[50]
*Catharsius molossus L.*	7.3; 6.5; 3.5; 2.0; 0.8–1.0	230; 195; 175; 145; 128; 113; 102.5; 55.7; 35.2	[15]

**Table 10 ijms-20-03943-t010:** py-GC-MS analysis of synthetic melanins.

Melanin Precursor	Pyrolysis Conditions	Py-GC-MS Results	Ref.
L-3,4 dihydroxyphenyIalanine (Dopa)	Curie-point 770 °C, 4 s	Main pyrolysis products: pyrrole, indole, catechol, indole and their derivatives Pyrrole derivatives more readily released during pyrolysis of oxidized melanin	[125]
		Main pyrolysis products: pyrrole, phenols, indoles, catechols, styrene and 2,3-dihydrobenzofuran Phenol, catechol and their alkyl derivatives, styrene and 2,3 dihydrobenzofuran—marker panel for unindolized DOPA-derived units Content of unindolized units in the polymer depends strongly on the oxidation time and DOPA concentration used during the melanin synthesis	[126]
	Curie-point 230 °C, 8 s	Main pyrolysis products: benzene, pyrrole, phenol, indole, and their alkyl derivatives	[132]
	Curie-point 770 °C, 8 s	Acidification with hydrochloric acid has no influence on DOPA-melanin pyrolysis products Dopa-metal complex melanin Additional pyrolysis compounds—pyrazines and pyridines and their alkylated derivatives Enhanced contribution of benzene Considerably reduced levels of pyrrole, phenol, benzenenitrile and indole-derivatives Dopa-Cu^2+−^melanins Increased 1,2-benzenediol content Dopa-Zn^2+−^ melanins Lower 1,2- benzenediol content	[128]
3,4 dihydroxyphenethyIamine (Dopamine)	Curie-point 770 °C, 4 s	Main pyrolysis products: pyrrole, indole, catechol, indole and their derivatives Pyrrole derivatives more readily released during pyrolysis of oxidized melanin Lower yields of indoles	[125]
		Free-copper dopamine melanin: high yields of phenol, catechol and their alkyl derivatives Copper-dopamine melanin: high content of indole compounds and small amounts of phenols and catechols	[127]
	Curie-point 770 °C, 8 s	Main pyrolysis product: 1,2-benzenediol Large quantities of pyrrole, phenol and their methyl derivatives Lower amount of indole	[129]
		Main pyrolysis products: 1,2-benzenediol, pyrrole, phenol and their alkyl derivatives Little amounts of benzenes and indoles Peroxynitrite treatment: Reduced level of 1,2-benzenediol Ratio of 1,2-benzenediol to phenol 1:2 (normally 2:1) Enhanced percentages of low molecular weight gases with low retention time	[130]
L-adrenaline	Curie-point 770 °C, 4 s	Free-copper adrenaline melanin Substantial amounts of unindolized precursor-derived units, e.g., catecholamines or their quinones. Copper-adrenaline melanin Mainly composed from indole type monomer units	[127]
L-tyrosine		Main pyrolysis products: pyrrole, indole, catechol, indole and their derivatives Pyrrole derivatives more readily released during pyrolysis of oxidized melanin Lower yields of indoles and higher amount of alkylindole derivatives	[125]
5-S-cysteinyldopamine	Curie-point 770 °C, 8 s	Benzothiazine-, benzothiazole- and thiazoloisoquinoline-derivatives—identified as panel markers	[129]
		Large quantities of benzothiazine-, benzothiazole- and thiazoloisoquinoline-deriva- tives Identification of 1H[1]benzothiopyrano[3,4-d]imidazol4-one as marker Peroxynitrite treatment: Benzothiazine- derivatives content reduced 2-fold Enhanced percentages of low molecular weight gases with low retention time	[130]
	Microfurnace 500 °C	Addition of benzothiazole and benzothiazine derivatives 7-methyl-5H-1,4-benzothiazin-5- one and 7-ethyl-2,3-dihydro-5H-1,4-benzothiazin-5-one to the list of pheomelanin type pigment marker panel	[133]
5-S cysteinylDOPA	Microfurnace 500 °C	Addition of benzothiazole and benzothiazine derivatives 7-methyl-5H-1,4-benzothiazin-5- one and 7-ethyl-2,3-dihydro-5H-1,4-benzothiazin-5-one to the list of pheomelanin type pigment marker panel	[133]
	Curie-point 230 °C, 8 s	Pyrolysate dominated by sulfur-containing compounds: thiophene, thiazole, and thiazolidine derivatives, hydroxybenzothiazole and 1,4-benzothiazine Pyrrole, indole, and pyridine derivatives also detected Thiazolidine derivative characteristic to 5-S cysteinylDOPA	[132]
Copolymers Dopamine:5-S-cysteinyldopamine (CysDA-melanin)	Curie-point 770 °C, 8 s	Main pyrolysis products: Benzothiazine-, benzothiazole-, thiazoloisoquinoline-derivatives, 1H[1]benzothiopyrano[3,4-d]imidazol4-one Little amounts of benzene, toluene, ethylbenzene, styrene, phenol, 4-methylphenol No 1,2-benzenediol, indoles	[129]
		Main pyrolysis products: Benzothiazine-, benzothiazole-, thiazoloisoquinoline-derivatives, 1H[1]benzothiopyrano[3,4-d]imidazol4-one Peroxynitrite treatment: Reduced percentages of all the typical pyrolysis products originated from 5-S- cysda-derived units Enhanced percentages of low molecular weight gases with low retention time	[130]

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
