# Peer review of "From Extraction to Advanced Analytical Methods: The Challenges of Melanin Analysis"

_ijms, 2019, doi:10.3390/ijms20163943_

Round 1
Reviewer 1 Report
This review summarizes the main literature focused on melanin analysis from the extraction methods to the advanced analytical techniques. Many tables are reported to compare results obtained from different papers. Table 4 reporting the AFM analysis of different types of melanins can be eliminated since only two cases are reported. The authors can explain these cases in the main text. In general, more figures have to be added. First, to help reader in following the topic a figure with Chemical Structures of the most important natural and synthetic melanins should be inserted. More some figures reporting also analytical results as an example for FTIR, NMR, Py-GC/MS, MALDI and LC-MS must be added and discussed. It is not clear for instance while in MALDI different molecular weights have been found. Try to give more analytical details and examples in each case since in this form the review looks like a simple list of papers. The review should be a bit more critical in advantages and drawbacks among the various extraction procedures and analytical techniques discussing which one could be more informative and helpful. A table illustrating also the type of samples (skin, brain, bacteria…) and the applied procedure can also be considered.Author Response
Dear Sir/Madam,
please find attached our responses to your comments and suggestions. Thank you for helping us to improve our manuscript.
The authors

Reviewer 2 Report
Pralea et al. summarize the extraction, purification, and characterization of ubiquitous biological pigments, melanins. The area is interesting, the manuscript could be publishable in the International Journal of Molecular Sciences after a major revision. I have the following comments and suggestions:
1. In the introduction section, I suggest to give a brief background of biological and commercial importance of melanins. This will justify why there is significant research interests in melanins.
2. I suggest to elaborate the “extraction and purification” section.
3. Lines 103 and 610, properties. Check spelling in other places too.
4. In section, 3.2, it would be helpful to include a representative TGA curve.
5. Line 145, is there any explanation why TEM studies are limited?
6. Line 190, check “the follow decrease”
7. Line 198, where.
8. What are R2 values in Fig 1b?
9. The review is more like a summary. I suggest that the authors critically analyze the literatare rather than just tabulating the studies. For example, the authors talk different techniques of characterization. There could be one of more limitations of these techniques. These need to be discussed.
10. It is is not very clear why it is challenging to analyze melanins and what are the outstandsing challenges and research questions. I also suggest to revise the title of the manuscript.
11. The manuscript requires minor edits in the language.
Author Response
Dear Sir/Madam,
please find attached our responses to your comments and suggestions. Thank you for helping us to improve our manuscript.
The authors

Round 2
Reviewer 2 Report
The authors revised the manuscript. I recommend for publication after minor edits.